# Importance of the 1+7 configuration of ribonucleoprotein complexes for influenza A virus genome packaging

Takeshi Noda[1,2,3,4,5], Shin Murakami[1,6], Sumiho Nakatsu[1], Hirotaka Imai[1,9], Yukiko Muramoto[1,4], Keiko Shindo[4], Hiroshi Sagara[7] & Yoshihiro Kawaoka[1,2,8]

The influenza A virus genome is composed of eight single-stranded negative-sense RNAs. Eight distinct viral RNA segments (vRNAs) are selectively packaged into progeny virions, with eight vRNAs in ribonucleoprotein complexes (RNPs) arranged in a specific "1+7" pattern, that is, one central RNP surrounded by seven RNPs. Here we report the genome packaging of an artificially generated seven-segment virus that lacks the hemagglutinin (HA) vRNA. Electron microscopy shows that, even in the presence of only seven vRNAs, the virions efficiently package eight RNPs arranged in the same "1+7" pattern as wild-type virions. Next-generation sequencing reveals that the virions specifically incorporate host-derived 18S and 28S ribosomal RNAs (rRNAs) seemingly as the eighth RNP in place of the HA vRNA. These findings highlight the importance of the assembly of eight RNPs into a specific "1+7" configuration for genome packaging in progeny virions and suggest a potential role for cellular RNAs in viral genome packaging.

[1] Division of Virology, Department of Microbiology and Immunology, Institute of Medical Science, University of Tokyo, 4-6-1 Shirokanedai, Minato-ku, Tokyo 108-8639, Japan. [2] International Research Center for Infectious Diseases, Institute of Medical Science, University of Tokyo, 4-6-1 Shirokanedai, Minato-ku, Tokyo 108-8639, Japan. [3] PRESTO Japan Science and Technology Agency, 4-1-8 Honcho, Kawaguchi, Saitama 332-0012 Japan. [4] Laboratory of Ultrastructural Virology, Institute for Frontier Life and Medical Sciences, Kyoto University, 53 Shogoin Kawahara-cho, Sakyo-ku, Kyoto 606-8507, Japan. [5] Laboratory of Ultrastructural Virology, Graduate School of Biostudies, Kyoto University, 53 Shogoin Kawahara-cho, Sakyo-ku, Kyoto 606-8507, Japan. [6] Department of Veterinary Microbiology, Graduate School of Agricultural and Life Sciences, The University of Tokyo, 1-1-1 Yayoi, Bunkyo-ku, Tokyo 113-8657, Japan. [7] Medical Proteomics Laboratory, Institute of Medical Science, University of Tokyo, 4-6-1 Shirokanedai, Minato-ku, Tokyo 108-8639, Japan. [8] Department of Pathological Sciences, School of Veterinary Medicine, University of Wisconsin-Madison, Madison, WI 53771, USA. [9] Present address: Department of Biological Informatics and Experimental Therapeutics, Graduate School of Medicine, Akita University, Akita 010-8543, Japan. Takeshi Noda and Shin Murakami contributed equally to this work. Correspondence and requests for materials should be addressed to T.N. (email: t-noda@infront.kyoto-u.ac.jp) or to Y.K. (email: kawaoka@ims.u-tokyo.ac.jp)

During repeated cycles of influenza virus replication, all eight viral RNA segments (vRNAs) must be incorporated into progeny virions for the virions to be infectious. Although eight distinct vRNAs are selectively incorporated into progeny virions[1–3], some studies have shown the existence of virions that lack one or more vRNAs and become infectious upon co-infection of complementary virions[4–6]. Reverse genetic studies have identified segment-specific cis-acting packaging signal sequences located at both the 3′ and 5′ ends of all vRNAs that drive their efficient incorporation into progeny virions[7–16]. Single-molecule fluorescent in situ hybridization analysis as well as a reverse genetics studies suggest that each progeny virion packages a single copy of each vRNA segment[17,18]. Electron microscopy (EM) analyses show that progeny virions incorporate eight ribonucleoprotein complexes (RNPs) arranged in a specific "1+7" pattern, with a central RNP surrounded by the remaining seven RNPs[5,19–22]. Taken together, these studies demonstrate that an influenza A virus tends to selectively package eight distinct RNPs arranged in a "1+7" pattern.

Interestingly, previous studies have experimentally demonstrated that seven-segment influenza A viruses lacking a certain vRNA could be generated by reverse genetics[7,15,23–25]. Although the growth of these seven-segment viruses is attenuated, the viruses can be stably passaged in cultured cells if the protein encoded by the omitted vRNA is provided in trans. The fact that authentic influenza A viruses package eight different RNPs in a "1 +7" pattern raises questions about the genome packaging of the seven-segment viruses, including how many RNPs are incorporated into the seven-segment virions and how are the RNPs arranged within those virions?

To answer these questions and to better understand the genome packaging mechanism of influenza A viruses, here we generated a seven-segment virus that lacks the HA vRNA and examined its genome packaging by using EM and next-generation sequencing (NGS). We demonstrate that the seven-segment virus packages host-derived rRNAs in place of the hemagglutinin (HA) vRNA, resulting in the packaging of eight RNPs arranged in the specific "1+7" pattern. Our findings suggest the importance of the "1+7" configuration of eight RNPs for the viral genome packaging.

## Results

**Generation of a seven-segment virus lacking HA vRNA.** We generated three different influenza A viruses based on A/WSN/33 (H1N1; WSN) by reverse genetics: the wild-type virus, a seven-segment virus that lacks HA vRNA (designated as the HA(−) virus), and an eight-segment virus that possesses a mutant HA vRNA (designated as the HAstop(+) virus). The mutant HA vRNA does not produce HA protein because of two in-frame stop codons introduced soon after the start codon in the HA mRNA (Fig. 1a). To generate the HA(−) and HAstop(+) viruses, 293T cells were transfected with seven plasmids for the expression of PB2, PB1, PA, NP, NA, M, and NS vRNAs or with eight plasmids for the expression of HAstop and the other seven vRNAs, together with five plasmids for the expression of the PB2, PB1, PA, NP, and HA proteins. HA vRNAs were undetectable in the supernatant of 293T cells producing the HA(−) virus, whereas HAstop vRNAs, in which no unexpected mutations were observed, were detected in the supernatant containing the HAstop(+) virus (Fig. 1b). Western blot analysis with an anti-HA antibody demonstrated that the HA protein was undetectable in Madin−Darby canine kidney (MDCK) cells infected with either the HA(−) or HA stop(+) virus (Fig. 1c).

Using MDCK cells stably expressing the HA protein (HA-MDCK cells), we compared the efficiency of infectious virus

production from plasmid-transfected 293T cells. Virus titers were determined by using plaque assays on HA-MDCK cells. The efficiency of HAstop(+) virus production in 293T cells was approximately 50 times higher than that of HA(−) virus production (Fig. 1d). We next assessed the growth properties of the HA(−) and HAstop(+) viruses in a multi-step replication cycle. HA-MDCK cells were infected with the viruses at a multiplicity of infection (MOI) of 0.01, and the virus titers were determined by using plaque assays on HA-MDCK cells. Although the growth of the HA(−) virus was significantly reduced compared with that of the HAstop(+) virus at each time point, the HA(−) virus showed relatively efficient replication, with titers reaching approximately $1 \times 10^7$ plaque-forming units/ml (Fig. 1e). These results suggest that the HA vRNA per se is important for efficient virus replication, but that it is not indispensable, consistent with previous reports[15,23].

**EM analysis of RNPs within the seven-segment virions.** To determine the number of RNPs packaged into the virions of the seven-segment virus, we prepared 100-nm-thick sections of HA (−), HAstop(+), and wild-type viruses budding from HA-MDCK cells and examined the RNPs within the respective virions by using EM as described elsewhere[5]. The exact number of RNPs within each virion cannot be counted by this method because the majority of virions examined here are not contained entirely within the 100-nm-thick ultrathin sections[5]; however, this approach should reveal the tendency of RNP incorporation into the virions of the respective viruses. In the HAstop(+) virions, eight RNPs arranged in the same pattern as in the wild-type virions were observed (Fig. 2a, b). Approximately 30% of the HAstop(+) virions contained eight RNPs arranged in the "1+7" pattern (Fig. 2e). HAstop(+) virions containing the eight RNPs were in the majority, but the proportion was slightly smaller than that of wild-type virions (Fig. 2d), probably due to mutations introduced into the 3′ packaging signal regions of the HA vRNAs. Surprisingly, HA(−) virions incorporated eight RNPs arranged in the specific "1+7" pattern with a similar frequency as the HAstop (+) virions; approximately 30% of the HA(−) virions examined contained eight RNPs (Fig. 2c, f). Virions incorporating 7, 6, 5, 4, 3, 2, and 1 RNP(s) were also observed, but those incorporating 8 RNPs were found most frequently (Fig. 2d–f, orange). Importantly, the proportion of virions containing seven RNPs was comparable between the eight-segment HAstop(+) virus and the seven-segment HA(−) virus (Fig. 2e, f, yellow).

To further confirm that HA(−) viruses package eight RNPs arranged in the "1+7" pattern, HA(−) virions, which were entirely embedded in 250-nm-thick sections, were subjected to scanning transmission electron tomography (ET) for three-dimensional (3D) reconstruction, and the number of RNPs within each whole virion was examined. Panels of digital slices of a whole virion from its top to the bottom, which were computationally generated from the 3D-reconstructed whole virion, are shown in Fig. 2g, h. As was reported for wild-type viruses[5,21], the eight RNPs of different lengths arranged in the characteristic pattern were clearly observed in the HA(−) virions (Fig. 2g; Supplementary Movie 1). Some HA(−) virions contained seven RNPs in a "1+6" arrangement (Fig. 2h; Supplementary Movie 2). Of the 12 HA(−) virions reconstructed in this study, 83% packaged eight RNPs, and the others packaged seven RNPs, whereas all of the wild-type virions examined in this study ($N = 11$) contained well-organized eight RNPs (Fig. 2i). Taken together, these results indicate that most budding progeny virions incorporate eight RNPs arranged in a "1+7" pattern, even in the presence of only seven distinct vRNAs.

**NGS analysis of the RNAs within the seven-segment virions.** Individual virions are known to package single copies of eight

distinct vRNAs; they usually do not package multiple copies of the same vRNA because of competitive inhibition by segment-specific packaging signal sequences[17]. Therefore, the question arises: what is the eighth RNP observed within the HA(−) virions? To identify the RNAs packaged into the HA(−) virions, we conducted NGS analysis. Wild-type and HA(−) viruses were grown in MDCK and HA-MDCK cells, respectively, and the supernatants containing the virions were treated with an RNase cocktail to remove RNAs outside of the virions. The virions were purified by continuous serial sucrose density gradient centrifugation, and the fractions containing virions were collected. Their purity was confirmed by use of sodium dodecyl sulfate-polyacrylamide gel electrophoresis (SDS-PAGE) followed by Oriole staining (Fig. 3a). Then the RNAs were extracted from the purified virions and subjected to NGS analysis. We obtained two datasets containing 70,281 and 90,911 reads corresponding to the wild-type and HA(−) virus RNAs, respectively. Each dataset was

mapped onto virus genome sequences and host genome sequences (i.e., dog genome sequences) (Table 1 and Fig. 3b). The plots for the PB2, PB1, PA, and HA vRNAs in both the wild-type and HA(−) virus datasets showed lower numbers of reads in the middle portion of the segments than those at both the 3′ and 5′ ends of the genome, implying the presence of defective interfering-like RNAs, as reported previously[26]. Although the number of NA vRNA reads was slightly higher in the HA(−) virus dataset than that in the wild-type virus dataset, there were no significant differences in the mapped read ratios of the respective vRNAs between the wild-type and HA(−) virus datasets. In the wild-type virus dataset, 97.6% of reads were mapped to the virus genome sequence, and very few reads were mapped onto rRNA sequences. Intriguingly, in the HA(−) virus dataset, 8.8 and 6.5% of reads mapped to 18S and 28S rRNAs, respectively. The sum of the 18S and 28S rRNA percentage of reads in the HA(−) virus dataset (15.3%) was comparable to the HA vRNA read percentage

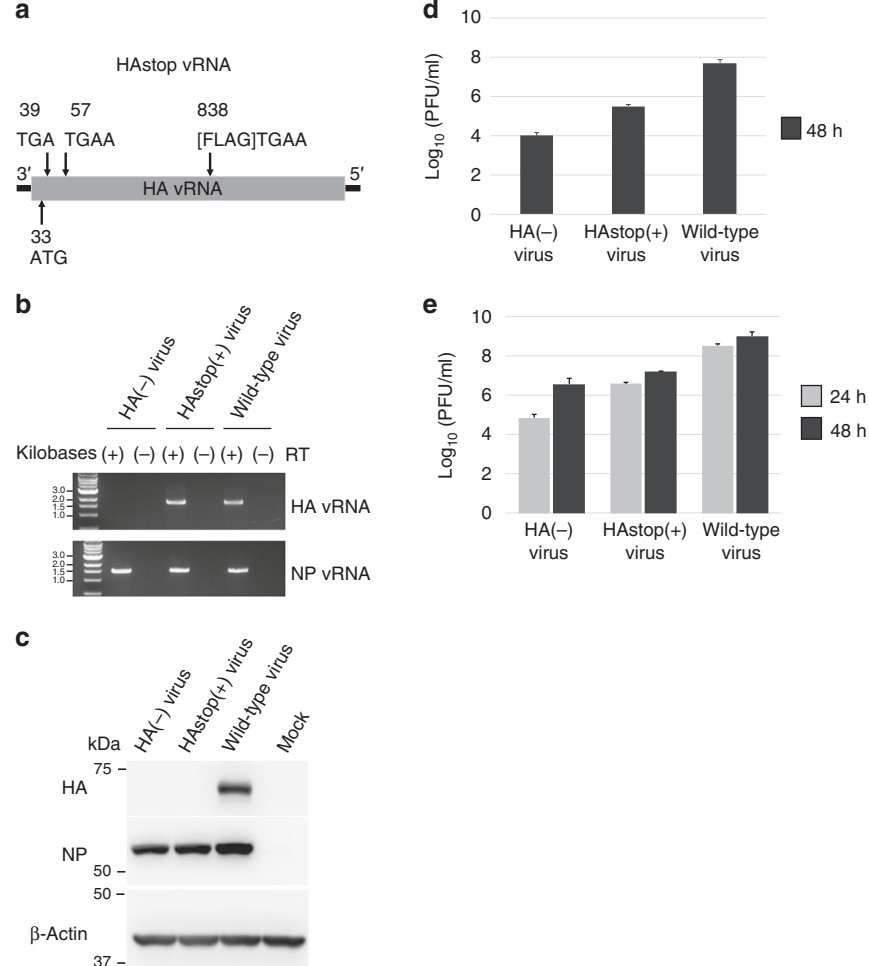

**Fig. 1** Generation of mutant influenza A viruses by use of reverse genetics. **a** Schematic diagram of a mutant HA vRNA (HAstop) that does not produce HA protein. It contains two stop codons and a FLAG epitope (Asp-Tyr-Lys-Asp-Asp-Asp-Asp-Lys) followed by a stop codon, in-frame with the HA reading frame. The numbers indicate the nucleotide positions from the 3′ end of the HA vRNA. Neither the HA protein nor the FLAG protein were detected by western blot analysis in cells transfected with this construct, using anti-HA and anti-FLAG antibodies. **b** RT-PCR analysis of the mutant viruses. The extracted vRNAs from the supernatants of plasmid-transfected cells were reverse-transcribed using a primer that was complementary to the conserved 12 nucleotides of the 3′ end of all eight vRNAs, followed by amplification by PCR with strand-specific primers for NP or HA vRNA. Reverse transcriptase (RT) reactions were performed with or without reverse transcriptase. **c** MDCK cells were infected with HA(−), HAstop(+), and wild-type viruses, respectively. At 12 h postinfection, the cell lysates were subjected to western blot analysis using anti-HA and anti-NP monoclonal antibodies. **d** 293T cells were transfected with plasmids to produce the HA(−), HAstop(+), and wild-type viruses. Virus yields were determined at 48 h post-transfection by use of plaque assays on HA-MDCK cells. The data represent the mean ± SD (n = 3). **e** HA-MDCK cells were infected with the respective viruses at an MOI of 0.01, and the culture media were harvested at 24 and 48 h postinfection. The virus titers were determined by use of plaque assays on HA-MDCK cells. The data represent the mean ± SD (n = 3)

in the wild-type virus dataset (12.1%), suggesting that the HA(−) virus incorporated these rRNAs instead of the omitted HA vRNA.

Next, to clarify whether the 18S and 28S rRNAs detected by NGS were packaged in their full-length forms, we performed a northern blot analysis of RNAs isolated from the wild-type and HA(−) viruses by using riboprobes that detect HA, NA, and NS vRNAs and 18S and 28S rRNAs (Fig. 3c). HA vRNA was not detected in the HA(−) virions, and the amounts of NA and NS vRNAs detected in the HA(−) virions were similar to those in the wild-type virions. Full-length 18S rRNA was detected in the HA(−) virions but not in the wild-type virions. In contrast, two forms of 28S rRNA of different lengths were detected in the HA(−) virions. A single band of about 2000 or 2500 nucleotides in length was detected with the riboprobes corresponding to nucleotide positions 1081−2042 or 3541−4679 of the 28S rRNA, respectively, suggesting that two fragments derived from the 28S rRNA were packaged into the HA(−) virions. This is consistent with our detection of two bands with the riboprobes corresponding to nucleotide positions 1081−4679 of the 28S rRNA. Accordingly, to

determine the respective sequences of the two truncated 28S rRNAs, we performed 5′ and 3′ rapid amplification of cDNA ends (RACE). The smaller fragment of the 28S rRNA was composed of nucleotide positions 1−2026 of dog 28S rRNA, and the larger fragment was comprised of nucleotide positions 2448−4760. These data demonstrate that full-length 18S rRNA and two different 28S rRNA fragments were incorporated into the seven-segment HA(−) virions.

Furthermore, to reveal whether the rRNAs are incorporated into virions along with ribosomal proteins, we performed western blot analysis by using anti-ribosomal proteins (Fig. 3d). Ribosomal proteins RPS3, RPS5, and RPL7, which are known to bind to 18S and 28S rRNAs[27,28], were not detected in either the wild-type or HA(−) virions, suggesting that 18S and 28S rRNAs do not exist in HA(−) virions as ribosomes.

**The rRNAs within the virions form RNP-like complexes.** To determine whether the rRNAs incorporated into HA(−) virions

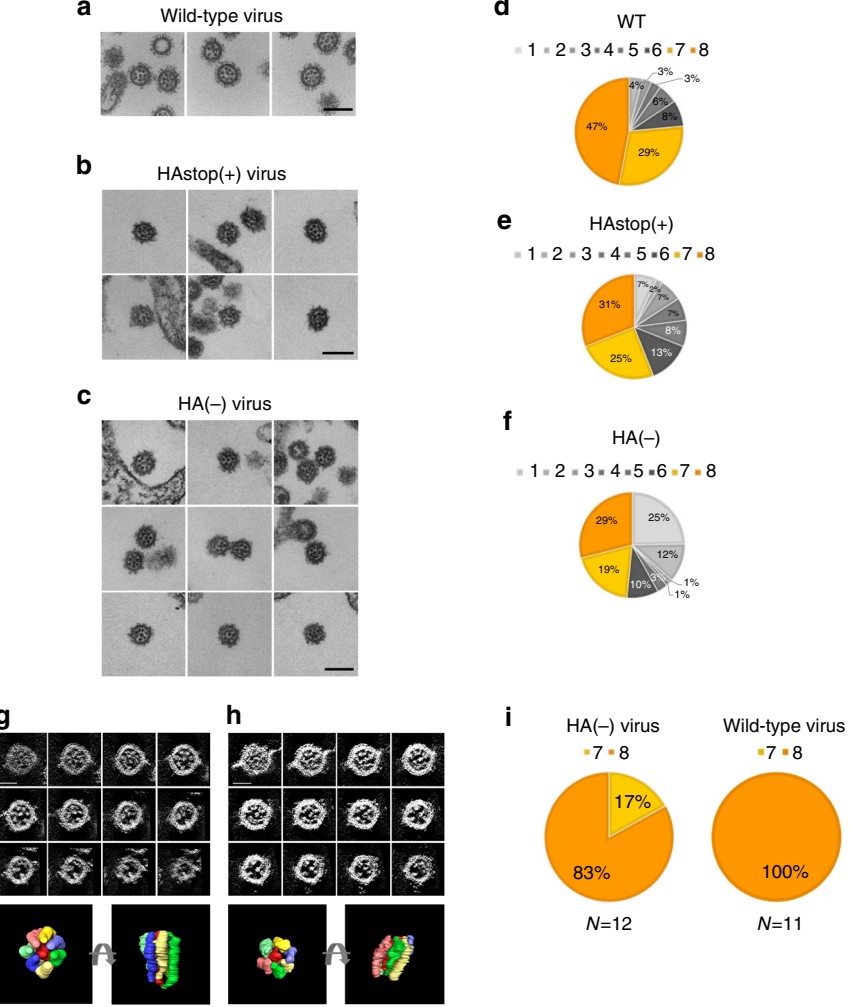

**Fig. 2** Electron micrographs of the HA(−), HAstop(+), and wild-type viruses. HA-MDCK cells were infected with the wild-type virus, HAstop(+) virus, or HA(−) virus, and ultrathin sections of budding virions were observed by EM. **a** Wild-type, **b** HAstop(+), and **c** HA(−) virions, which contain eight RNPs in the specific "1+7" configuration, bud from HA-MDCK cells. Bars, 100 nm. **d−f** Proportion of wild-type, HAstop(+), and HA(−) virions containing 8, 7, 6, 5, 4, 3, 2, and 1 RNPs. Approximately 100 virions embedded in 100-nm-thick ultrathin sections were counted for each virus. **g**, **h**. Digital sections of a whole HA(−) virion containing eight RNPs (**g**) and a whole HA(−) virion containing seven RNPs (**h**) from the top (upper left panel) to the bottom (lower right panel), computationally generated by electron tomography (ET). Three-dimensional models of the RNPs within the virions from the top (left) and side (right) views are shown below the digital sections panel. Bars, 50 nm. **i** The proportion of whole virions containing different numbers of RNPs. Approximately 10 virions each were 3D-reconstructed by ET, and the number of RNPs within each whole virion was counted

were associated with NPs and thus existed in RNP-like complexes, wild-type and HA(−) virions were treated with a nonionic detergent, and the RNPs inside the virions were isolated by means of glycerol density gradient ultracentrifugation. Each fraction was then collected and subjected to SDS-PAGE followed by Oriole staining. Membrane-associated HA and M1 protein fractions

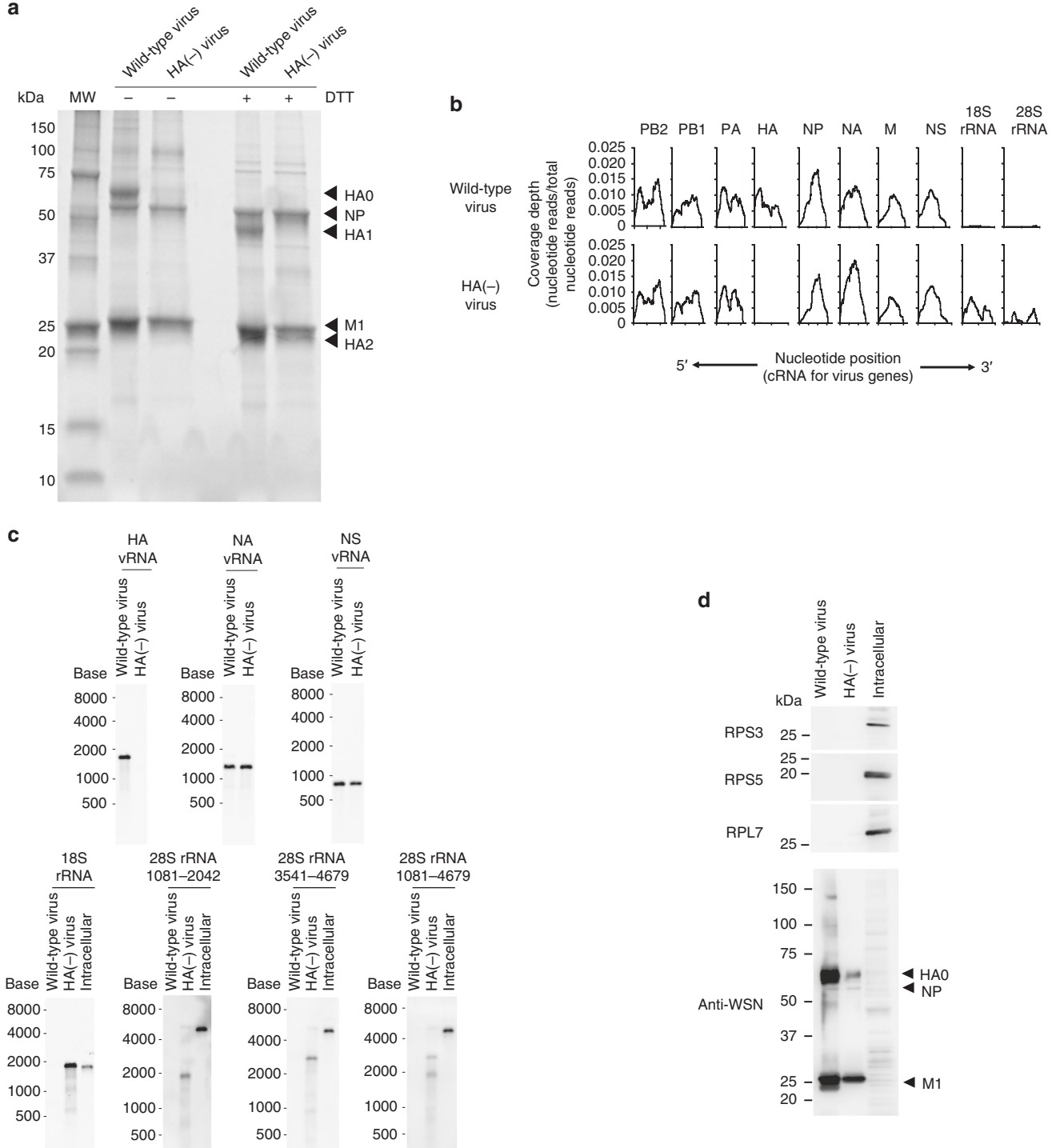

**Fig. 3** Next-generation sequencing (NGS) analysis of RNAs present within wild-type or HA(−)virions. **a** SDS-PAGE of purified wild-type and HA(−) virions treated with or without dithiothreitol (DTT) followed by Oriole staining. MW molecular weight marker. **b** Read coverage over the influenza virus genome. RNAs were extracted from purified virions and subjected to NGS analysis. Read sets from the wild-type and HA(−) viruses were mapped onto the influenza A virus genome and 18S and 28S rRNA sequences. Coverage depth is indicated as a ratio (coverage depth per nucleotide/total nucleotides read). The read sets of wild-type and HA(−) viruses are shown in the upper and lower columns, respectively. Nucleotide positions are indicated in the cRNA-sense for each vRNA. **c** An equal amount (5 ng) of RNA was subjected to northern blot analysis. The HA, NA, and NS vRNAs, 18S rRNA, and three different 28S rRNA (representing nucleotide positions 1081–2042, 3541–4679, and 1081–4679) specific riboprobes were used for detection. Intracellular RNAs from uninfected MDCK cells were used as a control. **d** Purified wild-type virions, HA(−) virions, and cell lysates were subjected to western blot analysis using anti-RPS3, anti-RPS5, and anti-RPSL7 antibodies and an anti-whole influenza virion polyclonal antibody

(fractions 2–4) and RNP fractions (fractions 6–7 for wild-type virus and 6–8 for HA(−) virus) were separated by density gradient ultracentrifugation (Fig. 4a, b). Undisrupted virions or partially disrupted virions sedimented in fractions 9–11. The amount of HA in the HA(−) virions, which originated from the HA from the HA-MDCK cells, was much lower than that in the wild-type virions. Then RNAs were extracted from each sample fraction, denatured, and separated on agarose-formaldehyde gels, which were subjected to northern blotting using riboprobes for influenza virus NS vRNA, 18S rRNA and 28S rRNA. NS vRNA, which was used as an example of vRNA, was detected in fractions 6 and 7 in the wild-type and in fractions 6–8 in the HA(−) viruses, respectively (Fig. 4c, d). The 18S and truncated-28S rRNAs were detected in fractions 6–8 where RNPs were detected in the HA(−) virus (Fig. 4f, h), suggesting that these rRNAs were likely present in RNPs. No 18S or 28S rRNA was found in the wild-type virus samples (Fig. 4e, g).

We further examined whether the 18S rRNA in the HA(−) virions was associated with NP protein by using an immunoprecipitation assay. Briefly, purified wild-type and HA(−) virions were disrupted by a nonionic detergent, and the lysates were incubated with an anti-NP antibody, followed by precipitation with Protein G beads. NP proteins (Fig. 5a) and NS vRNA (Fig. 5b), which was used as an example of vRNA, were co-immunoprecipitated from wild-type and HA(−) virions, suggesting that the anti-NP antibody immunoprecipitated RNP complexes. Within the precipitates, 18S and truncated-28S rRNAs were detected in HA(−), but not in wild-type virions (Fig. 5c, d), indicating that the 18S and truncated-28S rRNAs were associated with NP molecules. Therefore, it is highly likely that these rRNAs were incorporated into the HA(−) virions and formed RNP-like structures.

## Discussion

Influenza viruses selectively package eight distinct vRNAs, which are present in RNPs and arranged in a specific "1+7" pattern in progeny virions. Here we have shown that a seven-segment virus that lacks HA vRNA, and thus possesses only seven distinct vRNAs, efficiently packages host-derived rRNAs as an eighth RNP, resulting in the incorporation of eight well-organized RNPs into progeny virions in the same fashion as wild-type viruses. These results suggest that the assembly of eight RNPs into the specific "1+7" configuration may serve some mechanistic purpose during the genome packaging of influenza A viruses.

Using NGS and northern blot analysis, we showed that a substantial amount of 18S and 28S rRNA was packaged into HA(−) virions but not into wild-type virions. Interestingly, the 18S rRNA found in the HA(−) virions was full length, whereas the 28S rRNA was divided into two fragments (2026 and 2313 bases in length, respectively) (Fig. 3c). This is probably because of a difference in the lengths of rRNAs; the nucleotide length of the 18S rRNA (1874 bases) is similar to that of the HA vRNA (1776 bases), but that of the 28S rRNA (4718 bases) is much longer than any of the influenza vRNAs. The molecular mechanism underlying the incorporation of the 18S and 28S rRNAs remains uncertain, because the 18S and 28S rRNAs do not contain sequences identical to the HA vRNA 3′ and 5′ ends, which contain promoter and segment-specific noncoding regions and are involved in genome incorporation[29,30], nor do they possess the packaging signal sequences of HA vRNA. However, it is important to note that a previous report showed that 54 nucleotides derived from host 28S rRNA were inserted into the HA vRNA of A/turkey/Oregon/71 (H7N3) virus after serial passages in chicken embryonic cells; most likely, genetic recombination between the HA vRNA and the 28S rRNA occurred during the genome replication[31]. In addition, influenza virus NP

has been shown to localize in the nucleolus, where ribosomal biogenesis occurs[11]. Nucleophosmin and nucleolin, which also localize in the nucleolus, relocalize from the nucleolus to the nucleoplasm and colocalize with NPs[32–34], suggesting the possibility that RNPs associate with ribosomal subunits, including rRNAs, during the course of infection. Our findings, together with these reports, suggest that rRNAs are in the vicinity of vRNAs during virus genome replication and are more prone than other cellular RNAs to be assembled into the "1+7" configuration as a substitute for missing vRNAs. Because other seven-segment viruses that lack PB2, PA, or NA vRNA segments have been generated[23,24], it would be of value to determine whether the virions of these viruses also incorporate eight well-organized RNPs and package 18S and 28S rRNAs in a manner similar to the HA(−) virions. Such studies would help us further uncover the molecular mechanisms of influenza virus genome packaging.

Ultrathin section EM of HA(−) virions clearly revealed that they packaged eight RNPs arranged in the characteristic "1+7" pattern and that the proportion of HA(−) virions containing seven RNPs was not greater than that of HAstop(+) virions (Fig. 2e, f). In addition, 3D analysis of HA(−) virions by ET confirmed that most of the HA(−) virions packaged eight RNPs (Fig. 2i). Although the packaging efficiency of the eight RNPs seems to be slightly lower for the HA(−) virus than for the wild-type virus, which robustly packages eight RNPs[5], our results strongly suggest that HA(−) virions prefer to package eight RNPs arranged in the "1+7" pattern, as the wild-type virions do. We cannot exclude the possibility that the HA(−) virions contain two copies of an RNP (i.e., two copies of a certain vRNA), and thus eight RNPs in total, because it has been reported that the influenza virus can package two copies of a certain vRNA, and thus nine vRNAs in total[35,36]. However, because vRNAs possessing identical packaging signals (e.g., defective-interfering RNA and its progenitor vRNA) specifically compete with each other[17,37–39], the incorporation efficiency of two copies of a certain vRNA into a virion would be low. Taken together, our findings suggest that the seven-segment HA(−) virions package rRNAs, in a bundle of eight RNPs, to substitute for the missing HA vRNA, suggesting that assembly of eight RNPs into the specific "1+7" arrangement facilitates efficient incorporation of RNPs into progeny virions.

Although HA(−) virions package eight RNPs as efficiently as HAstop(+) virions, HA(−) viruses did not grow as well as HAstop(+) viruses in HA-MDCK cells, with an HA(−) virus titer approximately 1 log lower than that of the HAstop(+) virus (Fig. 1e), which has been reported previously[15]. One possible explanation for this difference in growth is the efficiency of virion

## Table 1 Packaged RNAs in wild-type and HA(−) virions

| Reference | Length | Number of reads (% of total reads) | |
|---|---|---|---|
| | | Wild-type virus | HA(−) virus |
| PB2 | 2341 | 13,535 (19.5) | 15,459 (17.0) |
| PB1 | 2341 | 9857 (14.2) | 12,608 (13.8) |
| PA | 2233 | 11,869 (17.1) | 14,524 (15.9) |
| HA | 1775 | 8399 (12.1) | 1 (0.0) |
| NP | 1565 | 9162 (13.2) | 9101 (10.0) |
| NA | 1409 | 7982 (11.5) | 14,719 (16.1) |
| M | 1027 | 3956 (5.7) | 4237 (4.6) |
| NS | 890 | 3887 (5.6) | 4964 (5.4) |
| 28S rRNA | 4752 | 12 (0.02) | 5888 (6.5) |
| 18S rRNA | 1838 | 15 (0.02) | 8005 (8.8) |
| Dog genome | | 18 (0.03) | 344 (0.4) |
| Unmapped | | 1589 (2.2) | 1061 (1.1) |

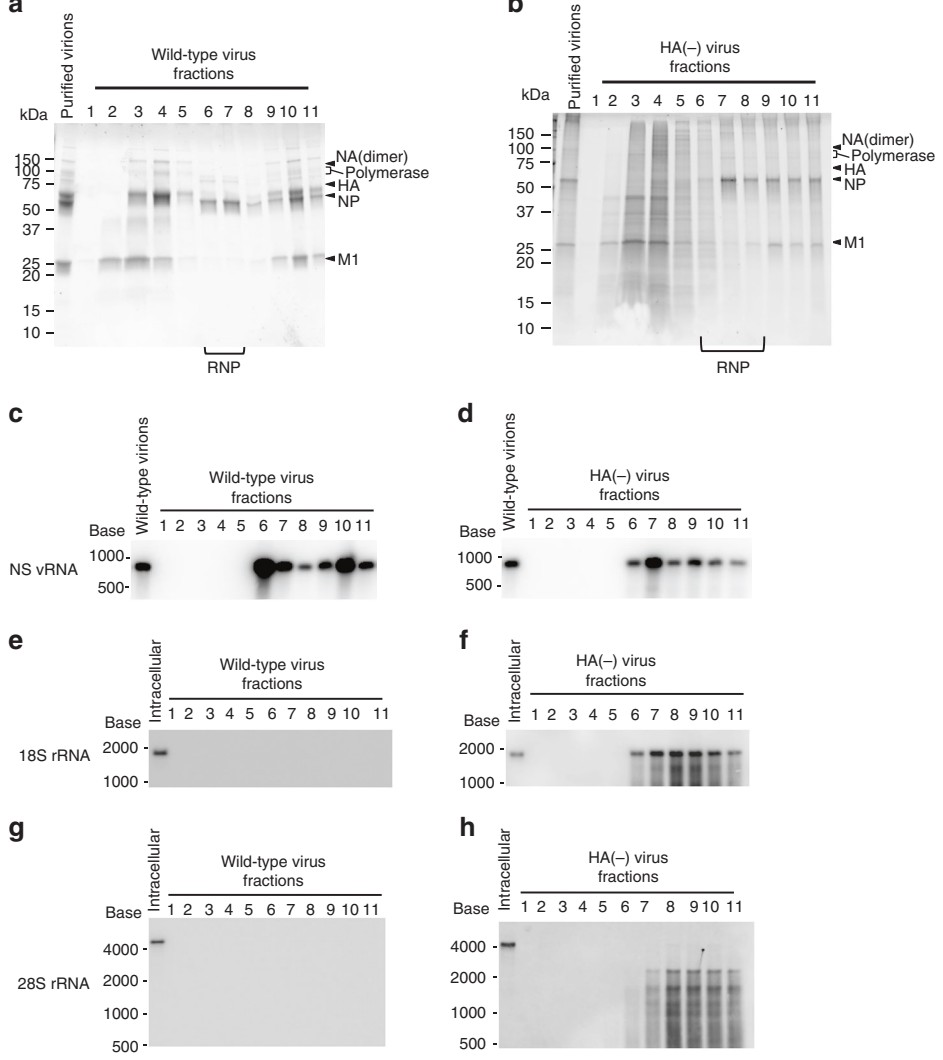

**Fig. 4** Fractionation of disrupted wild-type and HA(−) virions by glycerol density gradient centrifugation. Purified wild-type virus (**a**, **c**, **e**, **g**) or HA(−) virus (**b**, **d**, **f**, **h**) was disrupted by TritonX-100. The samples were subjected to 20–60% glycerol density gradient centrifugation. Eleven fractions were collected from the top of the gradient and numbered in ascending order from the top of the gradient to the bottom. **a**, **b** Each fraction was subjected to SDS-PAGE, followed by Oriole staining. **c–h** The presence of NS vRNA (**c**, **d**) 18S rRNA (**e**, **f**), and 28S rRNA (**g**, **h**) was examined by northern blot analysis using a digoxigenin-labeled oligonucleotide probe that bound to NS vRNA, 18S rRNA, or 28S rRNA (nucleotide positions 1081–4679). The numbers at the top of the panel denote fraction numbers

budding. Fujii et al.[7] reported that the presence of eight distinct vRNAs is favorable for efficient virion budding and that the number of budding virions with eight vRNA segments is approximately 10-fold higher than that of seven-segment virions lacking the HA vRNA. Therefore, a reduction in the virion budding efficacy of the HA(−) virus could lead directly to the reduction in growth.

In conclusion, we showed that influenza viruses package eight RNPs in a "1+7" pattern, even in the presence of only seven vRNAs. Our results highlight the importance of the assembly of eight RNPs for the genome packaging of this virus. The mechanisms by which the eight RNPs are organized into the characteristic configuration remain elusive; the packaging signal sequences that have already been identified on each vRNA segment, as well as potential packaging signal-like sequences that may exist on rRNA sequences, may be involved[10,30,40,41]. More detailed studies on the machinery responsible for assembly and uptake of the eight RNPs into virus particles are required to fully understand the genome packaging mechanism and influenza virus replication cycle.

## Methods

**Cells.** Human embryonic kidney (293T) cells (ATCC, CRL-3216) were maintained in Dulbecco's modified Eagle medium supplemented with 10% fetal calf serum and antibiotics. MDCK.2 cells (ATCC, CRL-2936) were maintained in minimal essential medium (MEM) containing 5% newborn calf serum and antibiotics. Both cells were grown in an incubator at 37 °C under 5% $CO_2$.

**Reverse genetics.** Reverse genetics was performed using plasmids (referred to as PolI plasmids) that contained cDNAs of the A/WSN/33(H1N1) viral genes between the human RNA polymerase I promoter and mouse RNA polymerase I terminator and eukaryotic protein expression plasmids under the control of the chicken β-actin promoter[42] as described previously[43]. Briefly, eight PolI plasmids and protein expression plasmids for PB2, PB1, PA, and NP were mixed with the transfection reagent TransIT-293 (Mirus, Madison, WI), incubated at room temperature for 15 min, and added to $10^6$ 293T cells cultured in Opti-MEM I (Invitrogen/Gibco, Carlsbad, CA). Forty-eight hours post-transfection, the supernatant containing viruses was harvested. To generate the mutant virus lacking the HA vRNA segment, seven PolI plasmids (all except for the PolI-HA plasmid) and five protein expression plasmids for PB2, PB1, PA, NP, and HA were employed. Forty-eight hours later, the supernatant was harvested and the mutant virus was propagated in HA-MDCK cells. Viral titers were determined by using plaque assays on HA-MDCK cells.

**Construction of the PolI HA plasmid.** To generate the PolI HAstop plasmid for the production of nonfunctional HA vRNA, we amplified the PolI-HA plasmid by

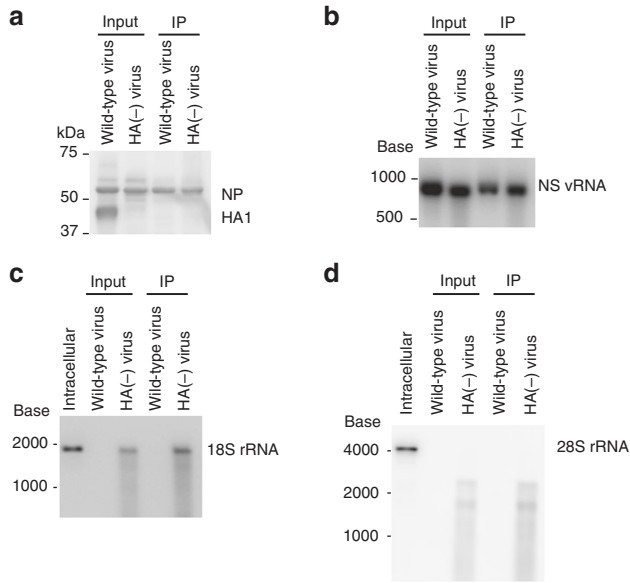

**Fig. 5** Co-immunoprecipitation of 18S and 28S rRNA with the NP protein. Purified virions were disrupted in buffer containing TritonX-100, incubated with an anti-NP antibody, and precipitated with Protein G beads. **a** The precipitates were subjected to western blot analysis using an anti-whole virion polyclonal antibody. **b–d** RNAs were extracted from the precipitate and were subjected to northern blotting by using riboprobes specific for NS vRNA (**b**), 18S rRNA (**c**), and 28S rRNA (**d**). IP immunoprecipitation

inverse PCR[44] using the back-to-back primers HA-stop-F (5′-CACACACGTCTCACTGGTCCTGTGAATATGCATTTGTAGCTACAGATGCAGACAC-3′) and HA-stop-R (5′-CACACACGTCTCACCAGTAGTTTTCACTTCATTTTGGTTGTTTTTATTTTCCCCTGC-3′). Then a sequence containing a FLAG epitope and stop codon (5′-CATGTGACTACAAGGACGACGACGACAAGTGAA-3′) was inserted into the NcoI site at nt 836, in-frame with the HA reading frame (Fig. 1a).

**RT-PCR**. The vRNAs were extracted from viruses released from plasmid-transfected 293T cells or from virus-infected HA-MDCK cells by using a QIAamp viral RNA Mini Kit (Qiagen) at 48 h post-transfection or 24 h postinfection. The extracted vRNAs were digested using an RNase-free DNase Set (Qiagen) according to the manufacturer's instructions to eliminate any residual transfected plasmid DNA. Each vRNA was reverse-transcribed using SuperScript III Reverse Transcriptase (Invitrogen, Eugene, OR) with a universal primer (a mixture of two primers, AGCAAAAGCAGG and AGCGAAAGCAGG) for all vRNA segments and then PCR-amplified using GoTaq Green Master Mix (Promega, Madison, WI) with strand-specific primers for either the HA (forward primer, AGCAAAAGCAGGGGGAAAATAAAAAC; reverse primer, AGTAGAAACAAGGGTGTTTTCCTT) or NP vRNA segment (forward primer, AGCAAAAGCAGGGTAGATAATCACTC; reverse primer, AGTAGAAACAAGGGTATTTTTCTTT).

**Viral growth kinetics**. HA-MDCK cells cultured in 12-well plates were infected with each virus at an MOI of 0.01 and incubated in MEM containing 0.3% bovine serum albumin at 37 °C. Aliquots of the supernatants were collected at 24 and 48 h postinfection and were titrated by using plaque assays on HA-MDCK cells.

**Electron microscopy**. Ultrathin section EM was performed as described previously[19]. Briefly, HA-MDCK cells were infected with the respective viruses at an MOI of 10, and the virus-infected cells were chemically fixed at 12 h postinfection. After dehydration with ethanol, embedding in Resin, and hardening at 60 °C, ultrathin sections were prepared and stained with uranyl acetate and lead citrate. To quantify the packaged RNPs in virions, 100-nm-thick sections were prepared, where most, but not all, of the RNPs present within the virions were expected to be countable considering the diameters of spherical influenza virions. The number of RNPs was determined by counting only those within transversely sectioned virions, because the RNPs within the longitudinally sectioned virions were uncountable as a result of overlapping images of RNPs within the virions. Empty virions were not considered, because the rate of empty virions is dependent on morphology; filamentous virions are more often empty when they are transversely sectioned[5]. For scanning transmission ET of virus-infected cells, semithin 250-nm-thick sections were stained with 2% uranyl acetate and Reynold's lead solution and were carbon-coated with the VE-2030 vacuum (Vacuum device, Japan). After plasma cleaning with a model 1020 plasma cleaner (Fischione, PA, USA), single-axis images of the

sections were acquired with a Tecnai F20 field-emission EM (FEI company, Netherlands) at 200 kV using an annular dark-field detector (Fischione, PA, USA). Images were collected with a 2cosθ° increment over a ±60° range with a pixel size ranging from 0.25 to 1 nm, and 3D structures were reconstructed by using the Inspect3D software (FEI company, Netherlands). Colored models of the RNP complexes within the virions were created with the Avizo 6.2 image processing package (Visualization Science Group, MA, USA).

**Virus purification**. The supernatants from virus-infected cells were collected at 36 h postinfection and centrifuged to remove cell debris. They were then treated with an RNase cocktail (RiboShredder, Epibio) for 12 h at 37 °C to remove RNAs on the outside of the virus particles. Virions in the supernatants were purified by ultracentrifugation on a continuous sucrose gradient consisting of 25–55% (w/v) sucrose in phosphate-buffered saline (PBS). After centrifugation at 142,000 × g for 3 h at 4 °C, fractions were collected from the top to the bottom. Virus-containing fractions were detected by using a hemagglutination assay. Fractions with peak HA titers were pelleted by ultracentrifugation at 142,000 × g for 3h at 4 °C. The pellets were resuspended in PBS and subjected to another round of continuous sucrose gradient centrifugation with the same conditions as those described above to further purify the virions. The purity of the virions was confirmed by SDS-PAGE analysis. The gels were stained with Oriole dye (Bio-Rad) and imaged using a GelDoc (Bio-Rad).

**Next-generation sequencing**. One hundred micrograms of RNA was prepared for NGS according to the cDNA Rapid Library Preparation Method Manual (Roche). Briefly, RNAs were fragmented in 0.1 M ZnCl$_2$ containing 0.1 M Tris buffer (pH 7.0), and RNAs <200 nt were removed using RNA-binding beads (Agencourt RNAClean XP; Beckman). Fragmented RNAs were used as templates to generate double-stranded cDNAs. The cDNAs were ligated with barcoded adapters. The prepared libraries were then clonally amplified by emulsion PCR using the emPCR Amplification Method Manual-Lib-L Kit (Roche) and sequenced on a GS Junior (Roche). Reads were mapped onto a viral genome (A/WSN/33 [Genbank accession numbers LC333182, LC333183, LC333184, LC333185, LC333186, LC333187, LC333188, and LC333189 for PB2, PB1, PA, HA, NP, NA, M, and NS vRNAs, respectively]) and mouse rRNA sequences (Genbank accession number X00686 for 18S; Genbank accession number X00525 for 28S) using the GS Read Mapper software (Roche). The nucleotide sequence data reported here are available in the DDBJ Sequenced Read Archive under the accession numbers DRX099540 and DRX099541.

**Northern blotting**. Total RNA was extracted from purified virions using the TRIzol reagent. Virion RNA (5 ng) was denatured and separated on 1.5 or 1.7% denaturing agarose-formaldehyde gels and transferred onto a nylon membrane with a molecular weight marker (BioDynamics Laboratory). Biotin-labeled strand-specific RNA probes for HA, NA, and NS vRNAs and 18S and 28S rRNAs were synthesized using a biotin-11-UTP (Roche) and a T7 RNA Expression Kit (Promega) according to the manufacturer's instructions. Northern blot analyses were performed using an ABC Kit (Vector) and a DIG block and wash buffer set (Roche) according to the manufacturer's instructions. The signals were detected with Clarity ECL substrate (Bio-Rad). The uncropped scans of the blots are shown in Supplementary Fig. 1.

**5′ and 3′ rapid amplification of cDNA ends**. The 5′ and 3′ ends of the 28S rRNA fragments that were incorporated into virions were determined by using the 5′ and 3′ RACE Kits (Roche), respectively. For 5′ RACE, 28S rRNAs were reverse-transcribed and then incubated in A-tailing buffer according to the instructions provided with the kit. For 3′ RACE, 10 ng of RNA extracted from purified HA(−) virions was polyadenylated by poly(A) polymerase (NEB). The 3′ end of the 28S rRNA was amplified according to the instructions provided with the kit. The amplified 5′ and 3′ ends of the 28S rRNAs were sequenced.

**Glycerol gradient centrifugation for RNP fractionation**. RNP fractionation was performed as previously described[45]. Briefly, purified virions were lysed for 1 h at 30 °C in a solution containing 50 mM Tris-HCl (pH 8.0), 100 mM KCl, 5 mM MgCl$_2$, 1 mM dithiothreitol (DTT), 2% lysolecithin, 2% Triton X-100, 5% glycerol, and 1 U/μl RNase inhibitor (Promega). The sample was then directly ultracentrifuged through a 30–70% glycerol gradient at 245,000 × g for 3 h at 4 °C, and 60 μl of each fraction was collected.

**Immunoprecipitation**. Purified virions were lysed for 1 h at 30 °C in a solution containing 50 mM Tris-HCl (pH 8.0), 100 mM KCl, 5 mM MgCl$_2$, 1 mM DTT, 2% lysolecithin, 2% Triton X-100, and 1 U/μl RNase inhibitor (Promega). The lysate was incubated with an anti-NP (2S347/3) mouse monoclonal antibody and Protein G Magnetic Beads (NEB). Precipitants were directly resuspended in TRIzol reagent, and RNAs were extracted for northern blot analyses. For western blot analyses, precipitants were directly dissolved in sample buffer.

**Western blotting**. For ribosomal protein detection, purified wild-type and HA(−) virions and MDCK cell lysates were subjected to SDS-PAGE under non-reducing conditions. Proteins were electroblotted onto polyvinylidene difluoride membranes (Millipore), which were then treated with Blocking One (Nacalai Tesque) for 1 h at room temperature and then incubated with anti-RPS3 rabbit polyclonal (Millipore, #ABE391, 1:1000 dilution), anti-RPS5 mouse monoclonal (Sigma, #WH0006193M2, 1:250 dilution), and anti-RPL7 rabbit polyclonal antibodies (abcam, #ab72,550, 1:2000 dilution) and anti-WSN rabbit polyclonal antibody (R309 prepared in house by using purified A/WSN virions as an immunogen, 1:2000 dilution) for 1 h at room temperature. After a 1-h incubation at room temperature with Sheep anti-mouse IgG secondary antibody for the mouse primary antibody and Goat anti-rabbit IgG secondary antibody for the rabbit primary antibodies, the blots were developed using a Chemi-Lumi One Super (Nacalai Tesque). For the detection of dissolved immunoprecipitants, the immunoprecipitants were boiled for 5 min without a reducing agent. Equal amounts of samples were subjected to SDS-PAGE. Proteins were electroblotted onto polyvinylidene difluoride membranes (Millipore), and after blocking with Blocking One (Nacalai Tesque) for 1 h at room temperature, the membranes were incubated with anti-WSN rabbit polyclonal antibody for 1 h at room temperature. After incubation with the Goat anti-rabbit IgG secondary antibody for 1 h at room temperature, the blots were developed using a Chemi-Lumi One Super (Nacalai Tesque). The uncropped scans of the blots are shown in Supplementary Fig. 1.

**Data availability**. The NGS data that support the findings of this study have been deposited in the DDBJ Sequenced Read Archive (accession numbers DRX099540 and DRX099541). All other data are available upon request from the corresponding authors.

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

## Acknowledgements

We thank Susan Watson for editing the manuscript. This work was supported by Exploratory Research for Advanced Technology (Japan Science and Technology Agency); Grants-in-Aid for Specially Promoted Research from the Ministries of Education,

Culture, Sport, Science, and Technology (MEXT) (grant numbers: 16H06429, 16K21723, and 16H06434); National Institute of Allergy and Infectious Disease Public Health Service research grants; Leading Advanced Projects (LEAP) for medical innovation from the Japan Agency for Medical Research and Development (AMED); and, in part, by the Global COE Program Center of Education and Research for Advanced Genome-Based Medicine for Personalized Medicine and the Control of Worldwide Infectious Diseases from MEXT. T.N. was supported by a Grant-in-Aid for Scientific Research (B) (grant numbers 25293107 and 16KT0111), a Grant-in-Aid for Challenging Exploratory Research (grant number 15K14853), a Grant-in-Aid for Scientific Research on Innovative Areas (grant number 15H01253), by PRESTO, Japan Science and Technology Agency (grant number JPMJPR13L9), by JSPS Core-to-Core Program A, the Advanced Research Networks, by a Grant for Joint Research Project of the Institute of Medical Science, University of Tokyo, by the Daiichi Sankyo Foundation of Life Science, by the Suzuken Memorial Foundation, by the Future Development Funding Program of Kyoto University Research Coordination Alliance, and by Setsubi Seibi Keihi of Kyoto University. S. M. was supported by a start-up Grant-in-Aid for Research Activity (grant number 24880014) and a Grant-in-Aid for the Encouragement of Young Scientists (B) (grant number 15K19104). S.N. was supported by a Grant-in-Aid from the Japan Society for the Promotion of Science.

## Author contributions

T.N. S.M., and Y.K. designed the experiments; T.N., S.M., S.N., H.I., Y.M., K.S., and H.S. performed the experiments; T.N. and Y.K. wrote the manuscript.

## Additional information

**Competing interests:** The authors declare no competing financial interests.

