## [Peer Review File · Nature Communications]

Reviewers' comments:

Reviewer #1 (Remarks to the Author):

An infectious influenza A virion selectively packages each of the 8 distinct RNPs arranged in a "1+7" pattern. However, 7-segment influenza A virus lacking a certain vRNA can be experimentally generated and stably passaged in cells so long as the omitted protein is provided in trans. The study by Noda et al. attempts to address a curious question about the genome packaging and arrangement of the 7-segment influenza A virus, using EM and next-generation sequencing. Their data convincingly demonstrate that the artificial 7-segment virions efficiently package host 18S and 28S rRNAs as the 8th vRNP to form the "1+7" pattern, suggesting that the assembly of 8 RNPs in the "1+7" pattern is important for influenza A virus packaging. The studies are carefully planned and well executed, with necessary controls. The data are novel and reveal important insights into influenza A virus genome packaging. That an infectious virus can package a host rRNA as one vRNP is also of general interests to a wide scientific community. Some minor comments are:

1. The Western Blot analysis of HA protein in the virus-infected MDCK cells were mentioned in the texts (page 7, lines 15-17) but not presented in Figures.
2. The % of 8-segmented virions in HA(-) virus budding from cells analyzed by EM is ~30% (Fig. 2d) but in the HA(-) virions analyzed by the scanning ET for 3D reconstruction is as high as 83% (Fig. 2g). It is unclear whether the HA(-) virions used in Fig. 2g have been purified through gradient ultracentrifugation and thus enriched for virions packaging 8 vRNPs. Please clarify.
3. Fig. 3a, how did the authors determine the identity of viral proteins on SDS-PAGE? The protein pattern of the purified A/WSN virions on SDS-PAGE differs from a previous publication (Shaw ML, Stone KL, Colangelo CM, Gulcicek EE, Palese P (2008) Cellular Proteins in Influenza Virus Particles. PLOS Pathogens 4(6): e1000085). Please clarify.
4. How many biological repeats were analyzed for NGS analysis of the purified virions?

Reviewer #2 (Remarks to the Author):

In the manuscript entitled "Importance of the 1+7 configuration of ribonucleoprotein complexes for influenza A virus genome packaging" the authors explore the assembly of influenza viruses lacking all 8 segments. Interestingly, they found that host ribosomal RNA is incorporated within an artificially generated seven-segment virus that lacks HA. For these studies, the authors used a seven-segment virus lacking HA vRNA and compared it to either WT WSN or a mutant virus that carries the HA vRNA but does not express the HA protein due to the incorporation of 2 STOP codons. EM and ET analysis was used to show that the HA(-) displays a 1+7 arrangement with the same frequency as the HAstop virus. They go on to determine that host ribosomal RNA coated with virus NP is being incorporated in virus particles that lack the HA vRNA. Surprisingly, they do not observe any host ribosomal RNA in WT virions.

Overall, this study is the first evidence of that host RNA can associate with viral NP and be packaged within an infectious virus. These observations are significant and exciting; however, there are few points that could strengthen the conclusions. The experiments were well controlled and the manuscript is well written.

Specific comments:

1. It is very surprising that the authors did not observe incorporation of any rRNA within WT viruses, especially given that ~70% of the viruses contained less than 8 segments. It seems that there may be an opportunity for host RNA to be packaged within the virions in the WT as well. Could the authors speculate as to why no rRNA is present in WT virions? In addition, what are the 2% of reads that are listed as 'other' in table 1?
2. The finding that NP is associated with packaged rRNA is also quite fascinating, and it is interesting to consider where this may happen during an infection given that rRNA is exclusively localized to the nucleolus. Could the authors determine whether the localization of rRNA is altered during infection or, at a minimum, expand upon the mechanism of rRNA incorporation within the discussion.
3. The authors should include analysis of ribosomal proteins within HA (-) virions, to strengthen the argument that the rRNA alone is packaged.
4. Given that the EM and ET studies were done at 36 hrs post infection, the replication study presented in Figure 1 should include a 36 hour timepoint. In addition, instead of a bar graph please represent the viruses as line graphs over time.
5. It is unclear how the delay in replication between the HA stop and WT virus is impacting the presence of the rRNA, the authors should include the titers of virus for each sample in figure 3.

Minor comments:

1. The authors need to clarify a few points within the materials and methods.
 - a. Specifically, regarding the virus purification protocol, it is unclear if the RNase treatment and gradient fractionation were performed only for the experiments presented in Figure 4 or for EM studies as well. The authors should clarify the preparation of virus used for each assay.
 - b. The authors should include a rationale for the extensive RNA treatment necessary prior to purification on the sucrose gradient. In addition, provide a rationale for why purification over two gradients was needed for these studies. Is this due to RNA on the outside of the virus particles?
 - c. Please include the number of fractions and the volume collected over the sucrose gradient presented in Figure 4.
2. The authors fail to mention the deposition information for the RNA seq data. It is standard that NGS data be deposited onto a common server, such as NCBI and the accession numbers corresponding to the data be published within the materials and methods.

Reviewer #3 (Remarks to the Author):

In this paper, Noda et al. characterized the arrangement of the ribonucleoprotein complexes (RNPs) in Influenza virions that lack the HA viral RNA (vRNA). By using electron microscopy they showed that these viruses incorporate 8 RNPs, which are arranged in the same 1+7 pattern as previously shown for WT virions. They then performed next generation sequencing and northern blot analysis, to show that the 8th RNA segment in the HA(-) virions corresponds to host-derived 18S and 28S ribosomal RNAs (rRNAs). Finally, they isolated the RNPs from the HA(-) virions and performed immunoprecipitation studies to show that the rRNAs are associated with the viral nucleoprotein, which combined with the electron microscopy studies strongly suggests that they are present in RNP-like complexes. The presented results are very convincing and are well discussed, and the paper is clearly written.

However, the authors do not provide a mechanism to explain this observation, and without it the manuscript's impact is greatly reduced. Unravelling such mechanism will certainly improve our understanding of Influenza virus morphogenesis. For example, without this mechanism is not easy to explain why the 18S and 28S rRNAs are not incorporated in WT viruses (Table 1). This fact suggests that the 18S and 28S rRNAs do not have any special properties that promote their packaging into virions, but still they are incorporated when the HA vRNA is not present. Furthermore, there is no good explanation for why specifically the 18S and 28S rRNAs are incorporated in HA(-) virions, and not any other cellular RNA.

Additionally, the authors should tone down some of their statements. For example, the authors state that the assembly of eight RNPs into the specific "1+7" arrangement is a critical step for incorporation of RNPs into progeny virions (page 15, lines 7-8 and page 17, lines 15 -16). According to their own results, this is not true, since 20% of A/WSN and 39% of A/PR virions do not incorporate 8 RNPs (Nakatsu et al, 2016, mBio). Furthermore, the authors do not really show that the 1+7 arrangement is strictly required, they just show that it is the most common arrangement. How can the authors be sure that the 1+7 arrangement is not just the result of packaging 8 "rods" into a confined space, such as an Influenza virion?

Specific comments:

- 1.- Page 6, lines 1-2: The authors state that "To answer these questions and further characterize the genome packaging mechanism of influenza A viruses". As mentioned above, the authors results do not characterize the packaging mechanism of influenza viruses, so this sentence should be rephrased.
- 2.- The description of the results shown in Figures 1C & D, and Figures 2C & D (pages 8 and 9) should include the results for WT viruses, rather than simply referencing previous publications. The total number of virions studied for Figures 2C & D should be stated.

Reviewers' comments:

Reviewer #1 (Remarks to the Author):

An infectious influenza A virion selectively packages each of the 8 distinct RNPs arranged in a “1+7” pattern. However, 7-segment influenza A virus lacking a certain vRNA can be experimentally generated and stably passaged in cells so long as the omitted protein is provided in trans. The study by Noda et al. attempts to address a curious question about the genome packaging and arrangement of the 7-segment influenza A virus, using EM and next-generation sequencing. Their data convincingly demonstrate that the artificial 7-segment virions efficiently package host 18S and 28S rRNAs as the 8th vRNP to form the “1+7” pattern, suggesting that the assembly of 8 RNPs in the “1+7” pattern is important for influenza A virus packaging. The studies are carefully planned and well executed, with necessary controls. The data are novel and reveal important insights into influenza A virus genome packaging. That an infectious virus can package a host rRNA as one vRNP is also of general interests to a wide scientific community. Some minor comments are:

1. The Western Blot analysis of HA protein in the virus-infected MDCK cells were mentioned in the texts (page 7, lines 15-17) but not presented in Figures.

In response to the reviewer’s comment, we have added the western blot data to Fig 1c.

2. The % of 8-segmented virions in HA(-) virus budding from cells analyzed by EM is ~30% (Fig. 2d) but in the HA(-) virions analyzed by the scanning ET for 3D reconstruction is as high as 83% (Fig. 2g). It is unclear whether the HA(-) virions used in Fig. 2g have been purified through gradient ultracentrifugation and thus enriched for virions packaging 8 vRNPs. Please clarify.

In both Figs. 2d and 2g, budding virions on virus-infected cells were analyzed. The difference in the percentage of HA(-) virions containing 8 vRNPs between those analyzed by EM and those analyzed by the scanning ET for 3D reconstruction originates from the sample preparation. In EM analysis, 100-nm-thick sections were prepared. Since the length of budding influenza virions is ~200 nm, the majority of virions are not entirely embedded in these 100-nm-thick sections. Therefore, for virions

containing 8 vRNPs, we may see less than 8 vRNPs unless the entire virions are captured in the 100-nm-thick sections. By contrast, for the scanning ET for 3D reconstruction, 250-nm-thick sections were prepared; therefore, the entire virion was captured in these sections for the majority of virions examined. We have added a paragraph to clarify this point (Page 9, lines 1–4).

3. Fig. 3a, how did the authors determine the identity of viral proteins on SDS-PAGE? The protein pattern of the purified A/WSN virions on SDS-PAGE differs from a previous publication (Shaw ML, Stone KL, Colangelo CM, Gulcicek EE, Palese P (2008) Cellular Proteins in Influenza Virus Particles. PLOS Pathogens 4(6): e1000085). Please clarify.

The migration pattern of NP and M1 in our analysis is comparable to that in the paper by Shaw et al. The only appreciable difference we see is the migration pattern of HA0; in Shaw et al., HA0 appears to migrate slower than HA0 in our analysis. We therefore performed this analysis in the presence or absence of DTT. As you see in Fig. 3a, in the presence of DTT, the band we refer to HA0 disappeared and two new bands that correspond to HA1 and HA2 emerged. We therefore believe that our identification of each viral band is correct.

4. How many biological repeats were analyzed for NGS analysis of the purified virions?

We performed the NGS experiment only once. However, the substantial amount of 18S and 28S rRNA was reproducibly detected in only the HA(-) virions by Northern blot analysis.

Reviewer #2 (Remarks to the Author):

In the manuscript entitled “Importance of the 1+7 configuration of ribonucleoprotein complexes for influenza A virus genome packaging” the authors explore the assembly of influenza viruses lacking all 8 segments. Interestingly, they found that host ribosomal RNA is incorporated within an artificially generated seven-segment virus that lacks HA. For these studies, the authors used a

seven-segment virus lacking HA vRNA and compared it to either WT WSN or a mutant virus that carries the HA vRNA but does not express the HA protein due to the incorporation of 2 STOP codons. EM and ET analysis was used to show that the HA(-) displays a 1+7 arrangement with the same frequency as the HAstop virus. They go on to determine that host ribosomal RNA coated with virus NP is being incorporated in virus particles that lack the HA vRNA. Surprisingly, they do not observe any host ribosomal RNA in WT virions.

Overall, this study is the first evidence of that host RNA can associate with viral NP and be packaged within an infectious virus. These observations are significant and exciting; however, there are few points that could strengthen the conclusions. The experiments were well controlled and the manuscript is well written.

Specific comments:

1. It is very surprising that the authors did not observe incorporation of any rRNA within WT viruses, especially given that ~70% of the viruses contained less than 8 segments. It seems that there may be an opportunity for host RNA to be packaged within the virions in the WT as well. Could the authors speculate as to why no rRNA is present in WT virions? In addition, what are the 2% of reads that are listed as 'other' in table 1?

There may be some misunderstanding regarding the percentage of WT virions containing 8 vRNPs. In EM analysis, 100-nm-thick sections were prepared. Since the length of budding influenza virions is ~200 nm, the majority of virions are not entirely embedded in these 100-nm-thick sections. Therefore, for virions containing 8 vRNPs, we may see less than 8 vRNPs unless the entire virions are captured in the 100-nm-thick sections. By contrast, for the scanning ET for 3D reconstruction, 250-nm-thick sections were prepared; therefore, the entire virion was captured in these sections in the majority of virions examined. By the latter method, 100% of the WT virions examined here contained eight RNPs. That is why it is not surprising that only a few rRNAs were detected in the WT virions, as shown in Table 1. Regarding the 2% of reads listed as "other" in Table 1, we have added some additional detail about these reads to Table 1, although some of them could not be mapped.

2. The finding that NP is associated with packaged rRNA is also quite fascinating, and it is interesting to consider where this may happen during an infection given that

rRNA is exclusively localized to the nucleolus. Could the authors determine whether the localization of rRNA is altered during infection or, at a minimum, expand upon the mechanism of rRNA incorporation within the discussion.

In response to the reviewer's comment, we now discuss possible mechanisms of rRNA incorporation in the revised manuscript (Page 17, lines 8–15).

3. The authors should include analysis of ribosomal proteins within HA (-) virions, to strengthen the argument that the rRNA alone is packaged.

In response to the reviewer's comment, we performed western blot analysis of HA(-) virions with antibodies against three rRNA-binding proteins (RPS3, RPS5, and RPL7) and confirmed that these proteins were not detected in HA(-) virions. We have added these data to Fig. 3d and describe the results on Page 13, lines 9–13.

4. Given that the EM and ET studies were done at 36 h post infection, the replication study presented in Figure 1 should include a 36-hour timepoint. In addition, instead of a bar graph please represent the viruses as line graphs over time.

We did the EM and ET experiments at 12 hpi after infection at an MOI of 10, whereas we examined growth properties at 24 and 48 hpi after infection at an MOI of 0.01. Because we cannot directly compare these data obtained under different experimental conditions, we believe that such time-course experiments are not essential. However, since the detailed experimental conditions for EM and ET were not described in the manuscript, we have now added this information (Page 23, lines 15–18).

5. It is unclear how the delay in replication between the HA stop and WT virus is impacting the presence of the rRNA, the authors should include the titers of virus for each sample in figure 3.

In response to the reviewer's comment, we have now included information about the virus titers of the WT, HAstop(+), and HA(-) viruses in Figs. 1d and e.

Minor comments:

1. The authors need to clarify a few points within the materials and methods.

a. Specifically, regarding the virus purification protocol, it is unclear if the RNase

treatment and gradient fractionation were performed only for the experiments presented in Figure 4 or for EM studies as well. The authors should clarify the preparation of virus used for each assay.

Because virus-infected cells were employed for EM and ET studies, RNase treatment and gradient fractionation were not performed for the EM and ET studies. To clarify this point, we have modified the EM section in the Materials and Methods (Page 23, lines 15–18).

b. The authors should include a rationale for the extensive RNA treatment necessary prior to purification on the sucrose gradient. In addition, provide a rationale for why purification over two gradients was needed for these studies. Is this due to RNA on the outside of the virus particles?

We performed extensive RNase treatment to remove RNAs outside of the virions and repeated the gradient ultracentrifugation to purify the virions as much as possible. In response to the reviewer's comment, we now explain why we did extensive RNase treatment and repeated ultracentrifugation in the revised manuscript (Page 25, lines 4–5 and 12).

c. Please include the number of fractions and the volume collected over the sucrose gradient presented in Figure 4.

In response to the reviewer's comment, we have added the RNP-containing fraction numbers to Fig. 4 and the information about the volume (60 ul) to the Materials and Methods section (Page 27, line 18).

2. The authors fail to mention the deposition information for the RNA seq data. It is standard that NGS data be deposited onto a common server, such as NCBI and the accession numbers corresponding to the data be published within the materials and methods.

We have now uploaded the NGS data as described on Page 26, lines 7–9.

Reviewer #3 (Remarks to the Author):

In this paper, Noda et al. characterized the arrangement of the ribonucleoprotein complexes (RNPs) in Influenza virions that lack the HA viral RNA (vRNA). By using electron microscopy they showed that these viruses incorporate 8 RNPs, which are arranged in the same 1+7 pattern as previously shown for WT virions. They then performed next generation sequencing and northern blot analysis, to show that the 8th RNA segment in the HA(-) virions corresponds to host-derived 18S and 28S ribosomal RNAs (rRNAs). Finally, they isolated the RNPs from the HA(-) virions and performed immunoprecipitation studies to show that the rRNAs are associated with the viral nucleoprotein, which combined with the electron microscopy studies strongly suggests that they are present in RNP-like complexes. The presented results are very convincing and are well discussed, and the paper is clearly written.

However, the authors do not provide a mechanism to explain this observation, and without it the manuscript's impact is greatly reduced. Unravelling such mechanism will certainly improve our understanding of Influenza virus morphogenesis. For example, without this mechanism is not easy to explain why the 18S and 28S rRNAs are not incorporated in WT viruses (Table 1). This fact suggests that the 18S and 28S rRNAs do not have any special properties that promote their packaging into virions, but still they are incorporated when the HA vRNA is not present. Furthermore, there is no good explanation for why specifically the 18S and 28S rRNAs are incorporated in HA(-) virions, and not any other cellular RNA.

In response to the reviewer's comment, we now discuss possible mechanisms of specific rRNA incorporation in the absence of HA vRNA (Page 17, lines 8–15).

Additionally, the authors should tone down some of their statements. For example, the authors state that the assembly of eight RNPs into the specific "1+7" arrangement is a critical step for incorporation of RNPs into progeny virions (page 15, lines 7-8 and page 17, lines 15 -16). According to their own results, this is not true, since 20% of A/WSN and 39% of A/PR virions do not incorporate 8 RNPs (Nakatsu et al, 2016, mBio). Furthermore, the authors do not really show that the 1+7 arrangement is strictly required, they just show that it is the most common arrangement. How can the authors be sure that the 1+7 arrangement is not just the result of packaging 8 "rods" into a confined space, such as an Influenza virion?

With all due respect, in our previous paper (Nakatsu et al., 2016, mBio), the electron tomographic analysis of A/WSN, which is also used in this study, demonstrated that 100% of A/WSN virions packaged eight RNPs arranged in the specific pattern. For other strains, it was demonstrated that at least 80% of virions packaged eight RNPs. Given that most virions package eight RNPs arranged in the specific pattern in the strains we tested, the packaging of eight RNPs into the specific arrangement likely has some biological significance. We, however, agree with the reviewer's point that we do not have evidence to show that the assembly of eight RNPs into the specific "1+7" arrangement is a critical step for incorporation of RNPs into progeny virions or is strictly required. We have, therefore, toned down our statement in response to the reviewer's comments (Page 16, lines 7–9; Page 19, lines 2–3).

Specific comments:

1.- Page 6, lines 1-2: The authors state that "To answer these questions and further characterize the genome packaging mechanism of influenza A viruses". As mentioned above, the authors results do not characterize the packaging mechanism of influenza viruses, so this sentence should be rephrased.

In response to the reviewer's comment, we have modified the sentence (Page 6, lines 1–2).

2.- The description of the results shown in Figures 1C & D, and Figures 2C & D (pages 8 and 9) should include the results for WT viruses, rather than simply referencing previous publications. The total number of virions studied for Figures 2C & D should be stated.

In response to the reviewer's comment, we have now included the results for the WT virus in Figs. 1 and 2. The total number of virions studied for Fig. 2 was about 100, which was described in the figure legend (Page 41, lines 11-12).

REVIEWERS' COMMENTS:

Reviewer #1 (Remarks to the Author):

The revised manuscript has satisfactorily addressed the previous critiques and is recommended for acceptance.

Reviewer #2 (Remarks to the Author):

The authors provided acceptable responses to this reviewer's comments of the original manuscript. Given the importance of the finding for influenza biology, these observations are deserving of publication.

As before, this paper is well written and the changes made to material and methods section provide adequate details for reproducibility of the results..

Reviewer #3 (Remarks to the Author):

Noda et al have correctly addressed the issues I raised in my initial revision. Additionally, while they have not unravelled the mechanism behind the incorporation of 8 RNPs in viruses with only 7 vRNAs, they provide a plausible explanation for it. As the authors suggest in their manuscript, it will indeed be interesting to that see if viruses that lack PB2, PA, or NA vRNA segments also package rRNAs.

Response to reviewers' comments

Reviewer #1 (Remarks to the Author):

The revised manuscript has satisfactorily addressed the previous critiques and is recommended for acceptance.

Thank you for sparing your precious time to review our manuscript.

Reviewer #2 (Remarks to the Author):

The authors provided acceptable responses to this reviewer's comments of the original manuscript. Given the importance of the finding for influenza biology, these observations are deserving of publication. As before, this paper is well written and the changes made to material and methods section provide adequate details for reproducibility of the results.

Thank you for sparing your precious time to review our manuscript.

Reviewer #3 (Remarks to the Author):

Noda et al have correctly addressed the issues I raised in my initial revision. Additionally, while they have not unravelled the mechanism behind the incorporation of 8 RNPs in viruses with only 7 vRNAs, they provide a plausible explanation for it. As the authors suggest in their manuscript, it will indeed be interesting to see if viruses that lack PB2, PA, or NA vRNA segments also package rRNAs.

Thank you for sparing your precious time for reviewing our manuscript. We have begun to investigate host RNAs packaged into other seven-segment viruses by NGS analysis.